# Intermittent Fasting: Potential Utility in the Treatment of Chronic Pain across the Clinical Spectrum

**DOI:** 10.3390/nu14122536

**Published:** 2022-06-18

**Authors:** Jesse P. Caron, Margaret Ann Kreher, Angela M. Mickle, Stanley Wu, Rene Przkora, Irene M. Estores, Kimberly T. Sibille

**Affiliations:** 1Pain TRAIL—Translational Research in Assessment & Intervention Lab, Department of Physical Medicine & Rehabilitation, College of Medicine, University of Florida, Gainesville, FL 32607, USA; jesse.caron@ufl.edu (J.P.C.); mkreher@ufl.edu (M.A.K.); angela.mickle@ufl.edu (A.M.M.); stanleywu155@ufl.edu (S.W.); rprzkora@anest.ufl.edu (R.P.); irene.estores@medicine.ufl.edu (I.M.E.); 2Department of Anesthesiology, Division of Pain Medicine, College of Medicine, University of Florida, Gainesville, FL 32610, USA

**Keywords:** chronic pain, intermittent fasting, non-invasive management, prehabilitation, rehabilitation

## Abstract

Dietary behavior can have a consequential and wide-ranging influence on human health. Intermittent fasting, which involves intermittent restriction in energy intake, has been shown to have beneficial cellular, physiological, and system-wide effects in animal and human studies. Despite the potential utility in preventing, slowing, and reversing disease processes, the clinical application of intermittent fasting remains limited. The health benefits associated with the simple implementation of a 12 to 16 h fast suggest a promising role in the treatment of chronic pain. A literature review was completed to characterize the physiologic benefits of intermittent fasting and to relate the evidence to the mechanisms underlying chronic pain. Research on different fasting regimens is outlined and an overview of research demonstrating the benefits of intermittent fasting across diverse health conditions is provided. Data on the physiologic effects of intermittent fasting are summarized. The physiology of different pain states is reviewed and the possible implications for intermittent fasting in the treatment of chronic pain through non-invasive management, prehabilitation, and rehabilitation following injury and invasive procedures are presented. Evidence indicates the potential utility of intermittent fasting in the comprehensive management of chronic pain and warrants further investigation.

## 1. Introduction

Evidence has persistently upheld that dietary interventions have a consequential impact on health [1,2]. While many dietary interventions have been promoted for weight loss purposes, intermittent fasting (IF) has gained increasing attention due to the observed protective neurocognitive, physiological, and cellular benefits [3,4,5,6]. Importantly, IF has high potential clinical utility: it is simple, practical, easy to implement, and offers a wide range of applicability. Despite its promising role in preventing, slowing, and reversing disease processes, clinical implementation of IF remains limited, possibly due to the heterogeneity of IF regimens that have been investigated. Most IF regimens align with one of the following categories described in Table 1.

IF produces several characteristic adaptations consistent across multiple different schedules, including lower blood glucose and insulin levels, improved insulin sensitivity, increased fatty acid mobilization and ketone body production, decreased leptin and insulin-like growth factor 1 (IGF-1), decreased markers of oxidative stress and inflammation, enhanced neuroplasticity and neurogenesis, stimulation of autophagy, and increased parasympathetic tone [9,11,12,20,21,22,23,24].

The distinctive benefits make IF applicable across a wide array of health conditions. IF has been shown as beneficial in the prevention and management of type 2 diabetes mellitus, metabolic syndrome and cardiovascular disease [25,26,27,28], and in the prevention and treatment of cancer [29,30,31,32,33,34,35,36,37,38]. IF has also been investigated in the slowing of several neurologic conditions including age-related cognitive decline, vascular and Alzheimer’s dementia, Parkinson's disease, and Huntington's disease [6,39]. Additionally, animal models have shown protective effects on the CNS preceding stroke, traumatic brain injury, and spinal cord injury [40,41,42]. A beneficial role for IF in the management of inflammatory and autoimmune diseases including rheumatoid arthritis, psoriasis, psoriatic arthritis, ankylosing spondylitis, multiple sclerosis, asthma, and type 1 diabetes is also indicated [9,43,44,45,46,47,48].

Many of the health conditions implicated in IF research are direct causes of, or comorbid with, chronic pain. The purpose of the review is to evaluate the potential relevance of IF in the treatment of chronic pain across the clinical spectrum. The review will: (1) provide an overview of the evidence of the benefits of IF by biological and physiological systems; (2) summarize the different pain states to be considered in the application of IF; and (3) discuss the potential utility of IF across three areas of pain management: non-invasive management, prehabilitation, and rehabilitation following injury and invasive procedures.

## 2. Mechanistic Benefits of Intermittent Fasting

IF has been shown to impact the biology and physiology of multiple systems. Studies span preclinical and clinical models, ranging across different ages and a wide range of health conditions [49]. Findings will be summarized across the following four systems: metabolic, cardiovascular, immune, and neurobiological.

### 2.1. Metabolic Functioning

Robust research has been completed in animal models showing the benefits of IF across multiple health indicators including improving insulin and glucose sensitivity [31,50,51], lowering blood pressure [51], decreasing body fat [51], improving fat metabolism [52], and reducing atherogenic lipids [13,51]. Additionally, multiple studies have shown microbial gut composition changes and increased anti-inflammatory effects following an IF diet [53,54,55]. In mice, time-restricted feeding influences metabolic regulation facilitating fat versus glucose expenditure. In the fasted state, the production of ketones and associated signaling molecules affect multiple major organ systems. Effects are believed to persist even after food ingestion, in turn affecting circadian biology and gastrointestinal microbiota [55]. Additionally, the combination of alternate-day fasting and high-intensity exercise in rat models has demonstrated a synergistic effect, promoting more mitochondria and enhanced oxidative efficiency in skeletal muscle, a decrease in brown adipose tissue, and an increase in skeletal muscle hypertrophy. The synergistic effect was also seen in enhanced glucose tolerance and reduced insulin levels [56].

Relationships between metabolism and IF have been broadly investigated [57,58,59,60]. Weight loss resulting from IF has been a particular area of interest [61,62,63]. In a study of normal, overweight, and obese humans, alternate-day fasting was effective at reducing body weight, body fat, total cholesterol, and triglycerides [64]. Additionally, human studies have shown IF to reduce insulin and glucose levels [8,11,12,59,60,65,66,67,68,69], insulin resistance and sensitivity [5,70], blood pressure [5,70], multiple lipid markers [70,71], and oxidative stress markers [5,12] across healthy, overweight, and obese adults. Although some studies have found no changes in glucose or insulin levels [9,59,60,61,72,73] across similar populations, studies of prediabetic adults found improvement in fasting glucose levels [5,10,59,60,74,75]. Alternate day calorie restriction has also been shown to improve asthma-related symptoms including pulmonary function in overweight patients within two weeks of diet initiation. Improved clinical findings were associated with decreased cholesterol, triglycerides, oxidative stress markers, and decreased inflammatory markers in this population [9]. One limitation is that many clinical studies have been limited by small sample sizes. Due to the various fasting methods and studies conducted in differing populations (i.e., healthy weight, overweight, obese, prediabetic, asthmatic), it is difficult to determine whether one fasting method is superior in mediating metabolic changes and whether these changes are more beneficial to one subset of individuals.

### 2.2. Cardiovascular Functioning

Significant cardioprotective effects of IF are well explored in preclinical models. Both calorie restriction and alternate-day fasting in rats mediated decreased bodyweight, heart rate, systolic and diastolic blood pressure, decreased sympathetic tone, and increased parasympathetic activity, which was maximized following one month of dietary changes [24]. A study conducted on elderly rats found that alternate-day fasting had a protective effect on cardiac fibrosis by inhibiting oxidative stress and NF-kB activation [76]. Additionally, initiating an IF diet prior to a myocardial infarction reduced cardiac myocyte hypertrophy, myocardial infarct size, and ventricular dilation [77,78]. Similar findings were demonstrated in mice models of myocardial infarction, with additional evidence that mice with impaired autophagy had worse myocardial damage, suggesting autophagy is an important mediator for cardioprotective responses [79]. Implementing an IF diet after myocardial infarction in rats also improved longevity and recovery of heart function [80]. Finally, by activating angiogenic and anti-apoptotic factors, IF promoted the long-term survival of rats after congestive heart failure [80].

Malinowski and colleagues summarized multiple clinical trials that assess cardiac risk factors affected by IF including lipid metabolism, inflammation, blood pressure, obesity, and glycemic profiles [81]. Many studies show a reduction in low-density lipoprotein (LDL), reduced total cholesterol levels, decreased inflammatory markers, decreased blood pressure, increased heart rate variability, weight loss, improved insulin sensitivity, and reduced cardiovascular disease risk [26,81,82,83]. Another study in adults on a fasting-mimicking diet found lowered blood pressure and a decrease in IGF-1 [18]. Similarly, a study on combined time-restricted feeding and resistance training in healthy older men demonstrated reductions in IGF-1 [84]. In adults with pre-hypertension, a water-only fasting regimen significantly decreased blood pressure which was maintained with a subsequent vegan diet [85,86]. In a study of young women following short-term fasting, during the luteal phase of menstruation, lower heart rate and cortisol levels were indicated suggesting a greater relaxation effect which may benefit menstrual symptoms [87].

### 2.3. Immune Functioning

Numerous animal studies have investigated IF regimens on immune system functioning [20,48,51,88]. Animal studies have shown similar results to human studies with a reduction in pro-inflammatory cytokines including TNF-α, IL-6, IL-1β, IL-12, and IFN-γ [20,48,89,90]. Fasting has been shown to reduce signaling of PKA, an important regulator of T cell activation downstream of IGF-1, in turn promoting activation of hematopoietic stem cell renewal and regeneration [91]. Additionally, multiple studies have shown IF or caloric restriction to reduce the number of circulating monocyte and lymphocytes in the periphery [92,93]. There are extensive effects of IF on immune cell function across multiple pathways [94,95,96,97]. Additionally, animal models of using IF in the treatment of autoimmune diseases have also been explored; however, they are more limited than human studies [47,48,98,99].

Clinical studies have shown caloric restriction in overweight or obese individuals resulted in a reduction of pro-inflammatory cytokines [92]. Additionally, healthy individuals undergoing religious fasting also show reduced levels of pro-inflammatory cytokines including C-reactive protein, tumor necrosis factor -α (TNF-α), interleukin-6 (IL-6), and IL-1β [84,100,101]. A meta-analysis by Wang et al. indicated IF significantly reduced C-reactive protein concentrations but not TNF-α or IL-6 [102]. Jordan et al. evaluated blood cells of healthy normal-weight adults before and after short-term fasting and reported a significant reduction in circulating monocytes, aligning with animal models [92]. Additionally, IF regimens have been tested as potentially therapeutic in multiple inflammatory and autoimmune diseases including asthma [9,103], rheumatoid arthritis [43,104,105,106], multiple sclerosis [47], and psoriasis [48,107].

### 2.4. Neurobiological Functioning

There are ample animal models investigating IF effects on neuronal protection, plasticity, and regeneration [108,109]. A focal cerebral ischemia model in rats has shown IF to induce neuroprotection due to anti-apoptotic factors [110]. In normal adult mouse brains, the Notch 1 signaling pathway is involved in neurogenesis through activation of downstream brain-derived neurotropic factor (BDNF) [22,111]. Multiple rodent model studies have found IF to induce BDNF [20,21,109,112] which promotes neuronal survival, plasticity, and stress resistance [113,114]. Another study by Manzanero and colleagues using a mouse model of ischemic stroke found increased neurogenesis and reduced leptin after a three-month intervention of IF, suggesting this as a potential mediator of neuroprotection [115]. Additionally, caloric restriction and IF stimulate autophagy [23,116]. Upregulation of autophagy enhances the functioning of multiple organ systems and provides significant neuroprotective effects [23,110]. In a mouse model, IF has also been associated with increased expression of myelin proteins improving peripheral nerve function through thicker myelin sheath, decreased basal lamina redundancy, and reduction of aberrant proliferation of Schwann cells [117]. Many of these studies were conducted in younger mice. Studies in older mice indicate that IF protects against ischemic injury following stroke compared to younger mice [20] and mixed results regarding IF on the reduction in β-amyloid [118]. Importantly, consistent preclinical findings show the role of IF in improving learning and memory [39,119,120,121].

Evidence suggests IF may improve cognition, sensory-motor function, and motor performance, which could be enhanced with vigorous exercise and may be beneficial in the treatment of neurological disorders [122]. There are only a few human studies assessing the neurological benefit of IF. Population groups include individuals with Huntington's disease (HD), Parkinson's disease (PD), and Alzheimer’s disease (AD) with a focus on ketogenic diets rather than IF. However, ketogenic diets are thought to mimic a fasting metabolic state and may be used to guide the potential therapeutic benefits of fasting diets [123]. In small studies, the ketogenic diet improved motor and non-motor symptoms in individuals with PD [124,125,126] and improved cognition in individuals with AD [127]. The ketogenic diet has also demonstrated efficacy in the prevention of migraine [128]. In individuals with epilepsy, IF has produced a therapeutic response [129].

Multiple studies have also documented fasting has mood-related benefits including increased feelings of well-being, focus, and occasionally euphoria [130]. Proposed mechanisms include modulation in neurotransmitter and glucocorticoid levels, improved sleep, formation of ketone bodies, and release of endogenous endorphins [131]. The effect of fasting on concentration and focus may be related to increases in serum neurotransmitters associated with elevated mood including norepinephrine, epinephrine, and dopamine [132]. Animal studies have shown fasting increases available serotonin and tryptophan, which may explain reports of well-being [133]. Furthermore, alterations in neurotransmitters and other centrally acting chemicals may implicate fasting in pain conditions with a “volume control” component such as fibromyalgia, temporomandibular joint disorder, IBS, headaches, osteoarthritis and other regional pain syndromes [134]. Lastly, fasting is associated with improvements in sleep quality and REM cycles, and fewer periodic limb movements and nocturnal awakenings, all factors that may contribute toward mood improvement associated with fasting [135].

## 3. Physiology of Pain States

Although features often overlap, clinical presentations of pain may be grouped as distinct states which interact with the body’s energetic, metabolic, and autonomic environments. Loosely grouped, six pain states will be described: mechanical pain, inflammatory pain, neuropathic pain, ischemic pain, visceral pain, and centralized pain. The physician’s task is to interpret these pain states to aid prognosis, etiology, and treatment, Table 2.

### 3.1. Mechanical Pain

Mechanical pain results from excess pressure or mechanical deformation on a muscle, bone, joint, or soft tissue, which triggers mechanical nociceptors to send signals to the brain. Transient mechanical pain may or may not involve tissue damage. Continued mechanical insult, however, such as the degeneration of articular cartilage in osteoarthritis, can cause anatomic abnormalities that result in chronic mechanical pain. Many forms of musculoskeletal pain, including low back pain due to herniated disc or myofascial pain, tendinopathy, muscle strain, and bone fracture, as well as incisions breaking the skin, represent mechanical pain [136].

### 3.2. Inflammatory Pain

Pain is one of the four cardinal features of inflammation [139] and inflammation is a hallmark characteristic across multiple pain conditions [158]. The inflammatory cascade results in the release of cytokines such as IL-1, IL-6, and TNF-alpha and soluble mediators including prostaglandins that sensitize type III and IV afferent nerve fibers to produce pain signals. Myelinated type III fibers conduct fast signals that result in the perception of sharp pain, while unmyelinated type IV fibers conduct slower signals, resulting in the perception of a widespread, burning sensation [140]. Inflammatory pain conditions with evidence for a potential therapeutic role for IF include rheumatoid arthritis, psoriatic arthritis, ankylosing spondylitis, and multiple sclerosis [46,60,139,140,141]. Environmental, genetic, psychological, and lifestyle factors can also contribute toward chronic inflammation [139,141,142,159].

For example, obesity contributes to adverse metabolic functioning that perpetuates inflammation resulting in (1) chronic secretion of inflammatory molecules, (2) reduction of metabolic flexibility and (3) abnormal tissue remodeling leading to ischemia [160]. Adipose tissue is capable of secreting inflammatory molecules that function in cell signaling and result in adaptive anabolic changes. However, in the setting of obesity, excess adiposity results in chronic secretion of inflammatory cytokines (IL-6, MCP1, DAG, TNFa, etc.) resulting in enduring, low-grade inflammation [161]. Furthermore, the chronically positive energy balance associated with excess adiposity creates a new homeostatic set point marked by insulin and catecholamine resistance. This effectively reduces metabolic flexibility favoring inflammatory rather than anti-inflammatory immune pathways [160]. Adipose tissue hypertrophy itself perpetuates inflammation via overcrowding phenomenon that results in mechanical compression of tissue resulting in hypoxia and cell death [162]. Lastly, hypertrophic adipocytes are also more prone to cell death and injury, providing another source of inflammation [160].

### 3.3. Neuropathic Pain

Both central and peripheral mechanisms contribute to neuropathic pain and include changes in peripheral ion channels, abnormalities in spinal inhibitory neurons, and dysfunction of noradrenergic inhibitory descending pathways [145,146]. Nervous tissue injury at the level of the periphery, spinal cord, or brain can induce neuroplastic changes leading to chronic neuropathic pain [145,146]. Neuropathic pain may or may not follow a dermatomal distribution and is reported as burning, stabbing, throbbing, and/or shooting in quality [147].

Approximately 6–8% of the general population experiences neuropathic pain, presenting with persistent pain, paresthesia, and/or hyperalgesia [146]. Neuropathic pain can be assessed by self-report scales, nerve function based on electrophysiological methods, a physical exam, and biomarkers including structural nerve proteins, IL-8, substance P, and nerve growth factor [145,148,163]. Medical conditions complicated by neuropathic pain include diabetic neuropathies, traumatic neuralgia, and lumbar and cervical radiculopathy [146,164]. Neuropathic pain can co-occur with inflammatory or mechanical pain [140] and typically responds to tricyclic antidepressants (TCAs), serotonin-norepinephrine reuptake inhibitors (SNRIs), and anti-epileptics [148], with limited benefit from opioid analgesics [165].

### 3.4. Ischemic Pain

Diminished or absent perfusion to tissue results in the accumulation of metabolites, including ATP and lactic acid, and the release of inflammatory cytokines that sensitize group III and IV afferent nociceptors [149,150]. During the restoration of blood flow to previously ischemic tissue, a local surge in pro-inflammatory reactive oxygen species stimulates afferent nociceptive fibers and contributes to the pain of ischemia-reperfusion injury [150,151,152]. Ischemia-reperfusion injury is a chief contributor to surgical complications and post-surgical pain. Sympathetic vasoconstriction due to stress during surgery exacerbates ischemia-reperfusion injury, increasing the risk of occlusive events that can result in ischemic pain [152]. Perhaps the most significant source of chronic ischemia-related pain-causing morbidity and mortality is atherosclerosis. Endothelial injury, local inflammation in the vessel wall, and deposition of atherogenic lipids cause vessel narrowing and possible thrombosis, resulting in reduced or absent blood flow to tissues. Pain manifests as myocardial angina or infarction, peripheral vascular disease, and mesenteric ischemia. Hypercoagulable states and vaso-occlusive disorders such as sickle cell disease are other significant sources of chronic ischemic pain.

### 3.5. Visceral Pain

Visceral pain originates from internal organs including the lower airways, heart, mesentery, and hollow organs of the gastrointestinal tract [154,155]. While several of the mechanisms of visceral pain overlap with the pain states described above, visceral pain is distinct in several ways. Not all viscera are sensitive to pain (such as most solid organs and lung parenchyma); injury is not necessary to produce visceral pain (such as intestinal distention without an injury causing severe pain), and overt visceral injury may not be accompanied by pain (as severing the intestine evokes no pain) [154]; it has a vague quality and is difficult to localize [156]; and it is often referred to somatically innervated structures [156]. Stimuli for visceral pain include ischemia and inflammation, as well as distention and chemical irritants that would not elicit pain in other body systems. Common conditions with visceral pain components include menstrual pain, constipation or bowel obstruction, irritable bowel syndrome, and the ischemic pain conditions of visceral organs (notably cardiac ischemia and mesenteric ischemia) [147].

### 3.6. Central Pain

Central pain results from a primary disturbance in pain processing in the central nervous system resulting in diffuse or regional hyperalgesia [134]. Central pain represents an injury to the pain sensor itself and can result in amplified nociception. Mechanisms of central pain are still being understood and include elevated underlying levels of pronociceptive neurotransmitters (substance P, glutamate) and reduced levels of neurotransmitters that inhibit pain (serotonin, norepinephrine, dopamine). The endocannabinoid system has been heavily implicated in the modulation of central pain processing [157]. While classic clinical examples of primarily central pain states include fibromyalgia and chronic pelvic pain, more prevalent conditions, such as osteoarthritis, are increasingly thought to have a significant central pain component [140].

More recent publications identify a specific form of central pain, nociplastic pain, as a distinct clinical syndrome of symptoms that is distinguished by its unpredictable, multifocal, widespread and intense presentation [166]. Nociplastic pain results from the direct augmentation of central pain processors and is closely linked to other central pathways that are implicated in mood, fatigue, sleep and memory [166]. The mechanisms of nociplastic pain are largely unknown but are being investigated in pain conditions such as fibromyalgia, chronic low back pain, and tension-type headache.

## 4. Intermittent Fasting in the Treatment of Chronic Pain

IF shows positive effects on metabolic, cardiovascular, inflammatory, and neurobiological functioning with health benefits ranging from weight loss to improved sleep, mood, and cognition [49,131,167]. Importantly, although the mechanistic contributors for the described pain states differ, IF appears to address relevant components for each and provides promise as a therapeutic tool for patients living with chronic pain [131,167].

In osteoarthritis, for example, IF reduces fasting insulin, a known contributor to cartilage degradation [137]. IF also promotes weight loss to reduce mechanical stress on joints [138] and leads to improved mood, which may augment participation in physical therapeutic interventions [131]. Furthermore, IF addresses the inflammatory component of osteoarthritis by reducing adipose tissue, which promotes a systemic inflammatory state that likely contributes to osteoarthritis pathology [144]. There is also considerable evidence that IF can reduce pathogenesis and symptoms of numerous other inflammatory pain disorders, including rheumatoid arthritis, psoriatic arthritis, and inflammatory bowel disease, through attenuation of inflammation and oxidative stress [53,89,90,91,143]. In trials of fasting for the treatment of rheumatoid arthritis, IF consistently led to reductions in markers of inflammation such as CRP and improvements in disease scores [43,104,105,106]. An additional benefit of IF in inflammatory diseases is the stimulation of autophagy [23,116], which regulates inflammation, suppresses pro-inflammatory cytokine secretion, and has been implicated in the treatment of inflammatory bowel disease [143].

Considering neuropathic etiologies of pain, IF may offer benefits through enhanced synaptic plasticity by way of increased BDNF expression [109,112,113,114], as well as improved peripheral nerve function through thickening of the myelin sheath and decreased aberrant Schwann cell proliferation [117]. Further, through improved insulin sensitivity and prevention of type 2 diabetes mellitus pathology, IF has a potential role in the prevention or slowing of diabetic neuropathy [27]. Although the long-term effects of IF on the development of diabetic complications in humans are not yet understood, animal models have proposed fasting may prevent the development of macrovascular complications of diabetes such as diabetic retinopathy by (1) altering the gut leading to increased generation of neuroprotective molecules such as tauroursodeoxycholate (2) decreasing production of neuro-inflammatory advanced glycation products and (3) enhancing autophagy decreasing oxidative stress that contributes to nerve damage [168].

IF may also play a significant role in the prevention and attenuation of ischemic pain through several mechanisms. IF has been shown to improve recovery and decrease vascular inflammation following ischemic injury [79,80,115,153], attenuate sympathetic vasoconstriction [24], and reduce the development of atherosclerosis through the improved metabolic profile, reduction in atherogenic lipids, and reduced oxidative stress [25,60,71]. Furthermore, greater low-frequency heart rate variability has been linked to lower pain sensitivity and higher thresholds for pain, suggesting a role of sub-optimal autonomic functioning in the pathogenesis of chronic pain. Heart rate variability has also been tied to emotionality, further connecting pain sensitivity to central homeostatic mechanisms [169]. Intermittent fasting has been shown to improve heart rate variability in both animals and humans [24,83].

In visceral pain conditions, IF may offer benefits by modulating the gut microbiota, promoting intestinal regeneration and decreasing symptoms of inflammation [53]. Additionally, IF increases parasympathetic tone which is one of the major determinants of gut motility. Increased heart rate variability is a clinical indicator of parasympathetic tone [170]. Increased parasympathetic tone may counteract gastrointestinal nociceptive pathways at the level of preganglionic neurons of the dorsal vagal motor nucleus [171]. Patients who underwent interventions to increase vagal tone (deep breathing exercises, etc.) demonstrated improved thresholds for bone pain, increased gastroduodenal motility index and more frequent antral contractions [171]. These effects not only support the potential role of IF in inflammatory bowel disease [53,143] but also warrant investigation in conditions related to gut dysmotility, such as chronic constipation or irritable bowel syndrome. More is needed to elucidate the role of parasympathetic tone and the potential influence of intermittent fasting on somatic pain sensitivity.

Finally, IF has the potential to alleviate symptoms in central pain syndromes such as fibromyalgia through mood enhancement [130]; release of norepinephrine, epinephrine, dopamine, and cortisol [131,133]; increased availability of tryptophan and serotonin [133]; synthesis of neurotrophic factors [131,132,133,134]; and improved sleep quality [135]. Fasting is implicated in these mechanisms through its diverse effects on central processing via regulation of neurotransmitter homeostasis, autonomic tone, the endogenous opiate pathways and BDNF-mediated neuroplasticity as discussed throughout this paper.

The potential application of IF in the treatment of chronic pain is addressed below in the context of non-invasive pain management, prehabilitation, and rehabilitation following invasive procedures.

### 4.1. Non-Invasive Pain Management

Although most outpatient chronic pain programs include nutrition education, few include weight loss interventions despite robust evidence supporting the benefits of weight loss on pain and functioning [172,173]. The reduced access to weight loss interventions may be due to the intensive resources needed for successful adherence to a prolonged low-calorie diet. Not only is there strong evidence demonstrating that IF offers numerous benefits extending beyond traditional diet programs, implementation is uncomplicated, incurs minimal expense, and is applicable across diverse populations.

A review of IF by de Cabo and Mattson (2019) provides a framework to assist clinicians with communication about different IF regimens. Specific to chronic pain, an IF pilot study was completed on middle and older adults with chronic knee pain. Based on three 16 h fasts over a ten-day period, study findings indicated that the regimen was feasible, acceptable, and without adverse events [174]. Given the consistent body of evidence regarding neuroplastic benefits, IF may also be effective as an adjuvant treatment [49,175]. Many outpatient chronic pain programs have an interdisciplinary structure and employ multiple modalities [176,177]. Pairing IF with other chronic pain treatment interventions such as pharmacotherapy, exercise, education, psychotherapy, acupuncture, and meditation amongst others might bolster the benefit of these interventions [175].

### 4.2. Pre and Post-Invasive Pain Management Interventions

Surgery contributes to major physiological disturbances affecting neurobiological functioning and multiple organ systems including cardiopulmonary, gastrointestinal, and immune. Excluding surgical or anesthetic complications, the “surgical stress” response is the underlying pathophysiology of these negative sequelae, which are associated with increased risk of morbidity and mortality and increased costs for the patient and society [178]. Chronic post-surgical pain is driven by continued afferent nerve fiber activation from direct tissue damage, peri-operative ischemia, nerve compression, aberrant healing, disrupted attenuation of pain signals once healing has occurred, or sequelae of surgical complications [179,180]. Risk factors for complications (e.g., an infection) include obesity, tobacco use, and perioperative hyperglycemia [153,163]. Perioperative insulin resistance and resulting hyperglycemia are also associated with surgical pain [181]. IF preceding surgery may optimize the system to better manage surgical stress. Following surgery, given the multi-system impact, IF may promote improved recovery.

### 4.3. Prehabilitation

The “surgical stress” response can trigger a cascade of negative events accumulating in adverse outcomes including death. The solution to this problem is the restoration of normal or improved physiology. Unfortunately, there is no single intervention to prevent or block the “surgical stress” response, several multimodal interventions have demonstrated promising results providing a solid foundation for future advancements. Nutritional regimens, exercise protocols, pain management strategies, and psychological support are just a few interventions to improve outcomes for surgical patients and many of them are part of the “Enhanced Recovery After Surgery” (ERAS) protocols. ERAS protocols are multimodal perioperative care pathways designed to achieve early recovery after surgical procedures by maintaining preoperative organ function and reducing the profound stress response following surgery [178,182,183].

IF is aligned with approaches to attenuate the surgical stress response as outlined in ERAS protocols. Preclinical studies of IF or dietary restrictions have demonstrated beneficial effects on outcomes, particularly related to attenuation of the oxidative stress response related to surgery [152]. In 298 morbidly obese bariatric surgery patients, caloric restriction preceding bariatric surgery reduced postoperative complications [184]. Preoperative weight loss interventions and dietary restrictions are associated with shorter hospital stays and reduced perioperative surgical risks [152,185]. IF protocols alone or with exercise have shown promising results on postoperative outcomes in colorectal surgery patients. In a meta-analysis of several studies totaling 914 patients undergoing colorectal surgery, nutritional prehabilitation alone or combined with an exercise program significantly decreased the length of hospital stay by two days in patients undergoing colorectal surgery. In addition, there was some evidence that multimodal prehabilitation accelerated the return to presurgical functional capacity [163].

### 4.4. Post-Procedure and Injury Rehabilitation

Similar to prehabilitation, IF provides promising results after an insult or as a postoperative intervention to improve outcomes. For example, postoperative IF suppressed oxidative stress and neuroinflammation induced by chronic cerebral ischemia in a rat model after vascular occlusion. In addition, IF animals had a significantly better cognitive function following the insult when compared to animals in the control group [186]. IF after stroke injury in mice resulted in favorable outcomes compared to mice fed ad libitum after the injury [115]. Neurogenesis and cell death were attenuated in mice undergoing IF, as was the infarct size compared to mice not undergoing IF. The study by Jeong et al. is also supportive of IF after neurological insult [187]. Dietary strategies in rats after spinal cord injury such as IF (every other day fasting) showed improved hindlimb motor function when compared to animals receiving a regular diet [187]. A systematic review by Sayadi et al. about the impact of IF and ketogenic changes on neuroprotection in animal models found potentially beneficial effects on neuroprotection and nerve fiber regeneration [188]. One disadvantage in many animal studies, pointed out by Sayadi et al., is the lack of standardization of the different types of IF throughout the various studies. Importantly, findings indicate the positive effects of IF after an insult occurred, indicating that IF is a promising intervention after a physiological disturbance such as surgery, trauma, or tissue hypoperfusion.

Wound healing following surgery, trauma, or a burn injury is critical. Preclinical studies demonstrate beneficial effects on wound healing [189]. Luo et al. showed that two periods of 24 h fasting prior to or following an injury were capable of accelerating wound repair and regeneration facilitated by the induction of pro-angiogenic factors. Improvement in glycemic profile mediated by IF may provide additional wound healing benefits. IF will most likely also have beneficial effects in so-called “post-insult” states as long as it does not negatively impact the nutritional balance to promote wound healing and recovery. Although major studies in humans are not published yet, one can certainly extrapolate that IF would likely have beneficial effects in humans in the post-procedure, postoperative period, after a stroke, or during rehabilitation after an illness.

## 5. Considerations and Contraindications

A few studies have demonstrated the practicality and benefit of implementing IF in the treatment of individuals with chronic pain. In a systematic review by Müller et al., of four controlled studies investigating fasting followed by a vegetarian diet, a pooled analysis showed significant long-term improvements in pain and severity of disease with fasting [43,104,105]. Another study compared the Mediterranean diet to 8-day modified fasting in individuals with fibromyalgia and rheumatoid arthritis. Although the sample sizes were small and improvements in fecal flora were not indicated, patterns of clinical improvements were observed in both groups with significance for the individuals with rheumatoid arthritis only [190]. Additionally, a pilot study demonstrated the feasibility and acceptability of three times a week 16 h fast in individuals with chronic musculoskeletal pain [174]. Further, limited fasting trials have been conducted in health conditions associated with chronic pain. One prospective interventional study in Germany showed that patients who participated in fasting protocols for at least 7 days during their hospital stay showed improvements with chief complaint and quality of life compared to placebo [131,191].

When choosing a fasting regimen to implement, there are several advantages and disadvantages to consider. The various IF regimens have been well-characterized in the literature and demonstrated feasibility and effectiveness in animals, humans, or both. The relative impact of a regimen on the desired outcome cannot be reliably assessed, as comparisons across regimens have not been studied. Evidence suggests time-restricted feeding, a minimal fasting period from 12 to 18 h in a 24 h day has benefits [5,13]. Thus, the decision on the IF regimen could be individualized based on patient preferences and tolerability. Time-restricted feeding, 5:2, and modified alternate-day fasting offer the benefits of easier transition into and out of the regimen, long-term consistency, and promotion of lifestyle adaptations [192].

On the other hand, periodic fasting may be preferred for patients who would benefit from a more rapidly induced clinical effect, for overall healthy patients who can tolerate a more extreme change for a limited period of time, or for those who prefer regular patterns of eating for the majority of the month. Notably, therapeutic periodic fasting in the form of 8–10 consecutive days of 200–500 kcal per day has been shown to be a safe treatment method with very high adherence rates in patients with chronic pain disorders [131]. Extended fasting may even be preferred over intermittent fasting due to hunger suppression over the extended period of fasting [131].

The optimal pain management regimen should address multiple contributing factors; thus, a multi-modal treatment plan and changes to improve overall health are vital for both the reduction and prevention of numerous types of pain. Intermittent fasting is one intervention that may be used in conjunction with, or as an alternative to, conventional medications, physical therapy, behavioral therapy, and alternative therapies with demonstrated safety and effectiveness. IF may not only reduce pain symptoms, it can benefit physical and mental functioning with a possible downstream influence on mobility, and quality of life.

Contraindications. It should be noted that prolonged fasting, for example, less than 25% calorie needs continuously over a period of more than one day should be avoided immediately prior to or following surgery to avoid malnourishment in at-risk patients including: the elderly, those already malnourished, bariatric patients, and individuals diagnosed with cancer as there is a catabolic response induced by the surgical trauma itself [193,194]. Decreased energy consumption during hospitalization is associated with increased length of hospital stays, rates of readmission, and mortality, especially in frail and malnourished patients [195,196]. Traditional preoperative fasting guidelines that keep patients fasting from midnight before surgery may be outdated [197,198]. Additionally, there is evidence to suggest that preoperative supplementation with carbohydrates up to two hours ahead of surgery improves insulin resistance after surgery [199].

Caution should also be taken regarding fasting during pregnancy due to potential risks for inadequate maternal weight gain and supply of essential nutrients for fetal development, fetal hypoglycemia and growth restriction, and inadequate CNS and organ development [200]. Maternal nutrition plays an essential role in placental and fetal development. Undernutrition and maternal ketosis may result in fetal growth restriction, developmental abnormalities, and significant morbidity [173,195,201,202]. Data from Ramadan fasting suggest, however, that time-limited restricted feeding in pregnancy may be feasible and safe [203,204,205], decrease the risk of gestational diabetes [206], and improve metabolic markers including high-density lipoprotein, insulin levels, and visceral fat distribution [207]. More research should be conducted to determine the potential role of a monitored IF regimen that could be beneficial for certain candidates and their babies.

## 6. Summary

IF demonstrates benefits across multiple physiological systems in animal and human studies. Although much research is in pre-clinical models, data suggest that IF decreases inflammatory markers, improves insulin sensitivity and metabolic profiles, reduces cardiovascular risk, enhances cognition and sensory-motor function, and relieves symptoms of neurological and mood-related disorders. The benefits of IF directly target a number of mechanisms associated with different pain states and chronic pain conditions. The clinical significance of IF in the treatment of chronic pain and pre or post-invasive interventions are missing. A 12 to 16 h IF regimen is simple to follow, easy to implement, cost-effective, and well-tolerated across diverse populations. Caution should be taken regarding fasting for high-risk individuals such as children, pregnant individuals, and the elderly. There is a need for clinical trials to be conducted in individuals with chronic pain implementing recognized fasting methods to promote improved interpretation, clear guidance for clinical application, and repeatability across studies.

## Figures and Tables

**Table 1 nutrients-14-02536-t001:** Description of common fasting regimens.

Fasting Type	Regimen	Description
**Intermittent Fasting**	Complete-alternate-day fasting [7,8]	No energy-containing foods or beverages on fasting days alternating with ad libitum intake on consumption days
	Modified-alternate day fasting [9,10]	20–40% of energy requirements consumed on fasting days alternating with ad libitum intake on consumption days
	5:2 [11,12]	Restriction to 25% or less of calorie requirements 2 days per week (consecutive or non-consecutive days) with ad libitum intake the remaining 5 days
	Time-restricted feeding * [5,13]	Ad libitum energy intake within a 6–12 h period, no energy-containing foods for the remaining 12–18 h in a 24 h period
**Periodic fasting**	Prolonged fasting [14,15]	2 to 21 days of very little or no energy intake (water-only fast) followed by a period of ad libitum intake
	Fasting-mimicking diet [16,17,18]	Low-calorie, low-sugar, low-protein, high-unsaturated fat diet (30–50% of energy requirements) for 4–7 consecutive days with ad libitum eating the rest of the month.

* Ramadan fasting (no energy consumption from dawn to sunset during the month of Ramadan) may be considered as a form of time-restricted feeding [19].

**Table 2 nutrients-14-02536-t002:** Definitions, mechanisms, and potential benefits of fasting.

Pain State	Definition	Relevant Mechanisms	Examples	Potential Benefits of Fasting
**Mechanical pain**	Pain resulting from abnormal stress on muscles, bones, joints, or soft tissue	Mechanical nociceptors triggered by excess pressure or mechanical deformation on a muscle, bone, joint, or soft tissue due to acute or cumulative trauma. May or may not involve tissue damage [136].	- Osteoarthritic pain caused by degeneration of articular cartilage - Mechanical low back pain due to herniated intervertebral disk - MSK injuries including muscle strain, tendinopathy, bone fracture - Incision breaking the skin surface	- Reduced fasting insulin, which promotes cartilage degradation [137]. - Promotes weight loss to reduce mechanical stress on joints and improve symptoms of osteoarthritis [138]. - Improved mood and weight loss for increased participation in physical therapy [131].
**Inflammatory pain**	Persistent or recurrent pain due to inappropriate activation of the inflammatory response	The inflammatory cascade includes release of chemical mediators including cytokines and prostaglandins that sensitize nociceptors [54,139,140,141,142].	- Rheumatoid arthritis, psoriatic arthritis, systemic lupus erythematosus, and other autoimmune diseases - Gout - Inflammatory bowel disease - Upregulated pain from inflammatory mediators released from adipose tissue in obesity	- Decreased oxidative stress and inflammation, including CRP levels [100,101,102]. - Stimulates autophagy [23,116], which regulates inflammation, suppresses pro-inflammatory cytokine secretion, and has been implicated in the treatment of inflammatory bowel disease [143]. - Reduced adipose tissue, which promotes a systemic inflammatory state that likely contributes to osteoarthritis [144].
**Neuropathic pain**	Pain resulting from a lesion in the somatosensory pathway	- Damage to the nervi nervorum in ascending pathway results in hypersensitivity of nociception. Associated with neurochemical biomarkers substance P, nerve growth factor, IL-8 [145,146,147,148].	- Diabetic neuropathy, lumbar and cervical radiculopathy, traumatic neuralgia - Peripheral and central sensitization is thought to contribute to osteoarthritic pain.	- Enhanced synaptic plasticity via increased BDNF [109,112,113,114]. - Increased myelin protein expression resulting in thicker myelin sheath and decreased aberrant Schwann cell proliferation, thus improving peripheral nerve function [117].
**Ischemic pain**	Pain resulting from diminished or absent perfusion to tissues	During ischemic injury, metabolites including ATP and lactic acid accumulate, inflammatory cytokines are released, and group III and IV nociceptive afferents are sensitized. Ischemia-reperfusion injury generates free radicals and ROS, and increased microvascular permeability during reperfusion allows inflammatory cell infiltration and release of pro-algesic cytokines including IL-1 and TNF-alpha [149,150,151,152].	- Post-surgical ischemia-reperfusion injury - Peripheral vascular disease - Sickle cell crisis - Myocardial infarction - Mesenteric ischemia	- Improved recovery following ischemic injury and decreased vascular inflammation [79,80,153]. - Attenuated sympathetic vasoconstriction [24]. - Reduced development of atherosclerosis through improved metabolic profile [13,60,71].
**Visceral pain**	Pain originating from visceral organs including lower airways, heart, mesentery, and hollow organs of the GI tract	Stimulated by chemical irritants, ischemia, distention, and inflammation. Tissue injury is not required. May be referred to somatically innervated structures corresponding to the spinal level of the affected visceral site [154,155,156].	- Pain associated with inflammatory bowel disease, constipation, bowel obstruction, irritable bowel syndrome - Menstrual pain	- Modulation of gut microbiota, promoting intestinal regeneration and decreasing symptoms of inflammation [53]. - Improves GI motility via increased parasympathetic tone [24].
**Central pain**	Disturbance in pain processing in the central nervous system resulting in diffuse or regional hyperalgesia	Elevated underlying levels of pronociceptive neurotransmitters (substance P, glutamate) and reduced levels of neurotransmitters that inhibit pain (serotonin, norepinephrine, dopamine). Implication of the endocannabinoid system, but not the endogenous opioid system (which is augmented in fibromyalgia) [134,157].	- Fibromyalgia - Irritable bowel syndrome - Tension headache - Idiopathic low back pain - Chronic pelvic pain - Myofascial pain syndrome	- Mood enhancement—specifically, increased feelings of well-being, focus, and euphoria [130]. - Release of norepinephrine, epinephrine, dopamine, and cortisol in the early phase of fasting with corresponding activation of stress resistance mechanisms [131,132]. - Increased availability of tryptophan and serotonin in the CNS [133]. - Ketosis-induced neurogenesis, neurotrophic factor synthesis, and expression of neurotransmitter receptors [131,132,133,134]. - Improved sleep quality [135].

## Data Availability

Not applicable.

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
