# Peer review of "Intermittent Fasting: Potential Utility in the Treatment of Chronic Pain across the Clinical Spectrum"

_nutrients, 2022, doi:10.3390/nu14122536_

Round 1
Reviewer 1 Report
General and specific comments to nutrients-171823 entitled „Intermittent fasting: Potential utility in the treatment of chronic pain across the clinical spectrum“ (Caron et al.).
This is an interesting narrative review on the potential utility of intermittent fasting on chronic pain.
Several points should be considered additionally.
- Line 72ff: The relationship between adverse metabolic functioning and the development of chronic pain should be briefly explained. E.g., for the reader, it is important to understand the link between obesity and inflammation, insulin resistance and chronic pain. On the other hand, the connection between glucose regulation alone and pain does not seem to be as clear as thought (Aström et al., Glucose regulation and pain in older people – the Helsinki Birth Cohort Study. Prim Care Diabetes 2021;15(3):561-6). Additionally, the specific neuroimmune mechanisms in chronic pain should be explained. It seems more logical to show the animal data first and then the human data.
- Line 105ff: The relationship between adverse cardiovascular functioning and the development of chronic pain should be briefly explained. E.g., this section could explain the importance of heart rate variability and the autonomic nervous system in chronic pain conditions. It seems more logical to show the animal data first and then the human data.
- Line 157ff: The relationship between adverse neurobiological functioning and the development of chronic pain should be briefly explained. E.g. line 169 to 182 could be set in the beginning of this section. In particular the concept of pain sensitization and nociplastic pain should be explained. It seems more logical to show the animal data first and then the human data.
- Line 163 to 168: In the context of the ketogenic diet its potential effect on migraine should be added (for example see Moskatel & Zhang. Migraine and diet: Updates in understanding. Curr Neurol and Neurosci Rep 2022; Caminha MC et al., Efficacy and tolerability of the ketogenic diet and its variations for preventing migraine in adolescents and adults: a systematic review. Nutr Rev 2021).
Author Response
Item # 1
We greatly appreciate the comments and suggestions, definite improvements.
The animal data now proceeds the clinical data for all systems reviewed.
We discuss the relationship between systems and pain states in various sections as noted below:
Section 3.2 Inflammatory pain - The neuroimmune mechanisms in chronic pain are expounded on Lines 222-227.
Section 4. Intermittent fasting in the treatment of chronic pain - A discussion of the relationship between adverse metabolic functioning and chronic pain was added after the overview of pain states, Lines 313-331.
Item # 2
Thank you for suggesting, cardiovascular functioning and pain are addressed as noted below and in the text, blue font.
The animal data now proceeds the clinical data.
Section 2.2 Cardiovascular functioning – Human, Lines 125-128.
In Section 4. Intermittent fasting in the treatment of chronic pain, Lines 344-354.
In Section 3.4. Ischemic pain – Lines 273- 279 – Further explanation of the role of atherosclerosis and ischemia in chronic pain was added.
Item #3
Neurobiological functioning and pain are addressed as noted below and in the text, blue font.
The animal data now proceeds the clinical data.
Section 3.3. Neuropathic pain – Mechanisms for the development of chronic pain due to neurologic dysfunction are discussed in the section on neuropathic pain, Lines 273- 279.
Section 3.6. Central Pain, a section on nociplastic pain has been added, Lines 305-311.
Section 4. Intermittent fasting in the treatment of chronic pain
Lines 332-343 – Further discussion was added relating the neurologic benefits of IF to pathways involved in neuropathic pain.
Item #4
Thank you for suggesting this addition.
In Section 2.4. Neurobiological functioning - Information on the ketogenic diet and migraines has been added, Lines 181-189.
Reviewer 2 Report
In this aticle, Caron et al. analysed the potential utility of intermittent fasting in the field of chronic pain.
Even if the article is well written and the topic is interesting, some points should be underlined:
- In the introductions the Authors defined intermittent and periodic fasting. However, in the review there is no mention of possible advantages obtained by these different approaches. I suggest to critically discuss possible advantages and disadvantages. There are differences between the two modalities?
- The principal aim of the article should be the possible role of intermittent fasting for the management of chronic pain. Even if some pain mechanisms are discussed, the possible role of fasting and pain management is not fully described. Excessive role was played by the description of cardiovascular, immunological and neurological effects of intermittent fasting. I suggest to add a critical discussion as a new paragraph to critically discuss literature findings and their possible role in pain management along with possible limitations of this approach. A critical discussion should be implemented between authors thoughs and literature findings.
Author Response
Thank you for the suggestions and important additions.
a) A critical discussion of the advantages and disadvantages of different approaches to IF are addressed in a new section, Section 5. Considerations and contraindications, Lines 476-516, noted in blue font.
b) The section on potential relevance of IF mechanisms in differing pain states has been further developed in Section 4. Intermittent fasting in the treatment of chronic pain, Lines 313-367, and in Table 2, noted in blue font.
Round 2
Reviewer 1 Report
The manuscript has been much improved. I have no further comments.
Reviewer 2 Report
The revised version of the manuscrip was significantly improved and it is now suitable for publication.